# Inequity of Access: Scoping the Barriers to Assisted Reproductive Technologies

**DOI:** 10.3390/pharmacy11010017

**Published:** 2023-01-16

**Authors:** Amanda Mackay, Selina Taylor, Beverley Glass

**Affiliations:** 1Pharmacy, College of Medicine and Dentistry, James Cook University, Douglas, Townsville, QLD 4811, Australia; 2Centre for Rural and Remote Health, James Cook University, Mount Isa, QLD 4825, Australia

**Keywords:** in vitro fertilisation, IVF, pharmacist role, rural and remote

## Abstract

Infertility impacts millions of people of reproductive age worldwide, with approximately 10–15% of couples affected. When infertility is present, there are many potential barriers to treatment, leading to inequity of access. Assisted reproductive technologies (ART) are the mainstay of medical treatment for infertility and include procedures such as in vitro fertilisation. This scoping review aims to explore the barriers to accessing assisted reproductive technologies to highlight a potential role for the pharmacist in addressing these barriers. Five databases, including CINAHL, Emcare, Medline, Scopus, and Web of Science, were searched using keywords that resulted in 19 studies that explored barriers to initially accessing or continuing ART. Studies identified more than one barrier to accessing ART, with the most mentioned barrier being the geographic location of the patient, with others themed as psychological, financial, minority groups, educational level, and the age of the patient. Recommendations were made to address barriers to accessing ART, which included changes to government regulations to increase health education and promotion of infertility. Pharmacists’ accessibility, even in geographically remote locations, places them in an ideal position to address many of the challenges experienced by people accessing infertility treatment to improve outcomes for these people.

## 1. Introduction

Humans have a right to reproduce, however when infertility is present there are many potential barriers to accessing treatment, leading to inequity of access. Medical methods of treating infertility are varied, with most coming under the umbrella term of “Assisted reproductive technology” (ART) which is an option for people who have difficulty conceiving to improve their chances of pregnancy success [1]. The most common ART procedure is in vitro fertilisation (IVF); however, the term ART includes other medical treatments or procedures that attempt to attain pregnancy [1,2]. In Australia, data from 2019 estimates that 4.9% of all women who gave birth received some form of ART [1].

A diagnosis of infertility can be associated with emotional, psychological, and social effects on the woman and those around them [3,4]. ART is not a guaranteed cure for infertility, with 28.4% of women who undertake ART achieving a clinical pregnancy, of which 66.7% result in a live birth [5]. Unlike the treatment of most other medical conditions, ART is not accessible to all individuals who are diagnosed as infertile [6]. The financial cost of ART to the patient is arguably the most well-known barrier to patients beginning ART. However, there are other barriers that exist which prevent patients from initially accessing and/or continuing ART. As ART is a specialist medical service, facilities are generally only located in metropolitan and large regional areas, with greater populated cities providing more choice of service [7,8,9,10]. ART comprises complex treatments and stressful procedures, including daily self-administered injections, blood samples, ultrasound, and laparoscopic surgery, and involves a multidisciplinary team of medical specialists, midwives, registered nurses, laboratory staff, sonographers, administrative staff, psychologists, and pharmacists [11,12]. Treatment outcomes are not always predictable, and as initial success rates are relatively low, it is common for a patient to undergo multiple attempts [11]. Anecdotal evidence of physical, financial, psychological, cultural, and emotional impacts of going through the ART journey, combined with the above potential barriers leads to an understanding that there is not equal access for patients with equal needs.

The aim of this scoping review is thus to determine the barriers to initiating or continuing Assisted Reproductive Technologies (ART). As medications are at the forefront of ART, this review also attempts to identify areas where barriers can be addressed from a pharmacy practice perspective to enhance the patient experience and act as enablers to initiating or continuing ART.

## 2. Methods

### 2.1. Literature Search

A scoping search of the literature was completed to identify studies discussing barriers to people accessing ART. The five databases were searched to identify relevant studies and included CINAHL, Emcare, Medline, Scopus, and Web of Science. The following terms were used for the search: ART, assisted reproduct* tech*, infertility, assisted conception, IVF, in vitro fertili*ation, test tube bab*, barrier*, access*, obstacle*, and challenge*.

### 2.2. Selection of Relevant Studies

Studies were firstly excluded based on the content of the title and abstract, with the remaining studies’ full text screened to identify if they complied with the inclusion criteria, which were studies discussing ART/IVF/infertility and barriers to accessing treatment (both initiating or continuing). Exclusion criteria included: studies about contraception, studies on the outcomes of ART (rather than barriers to accessing), oncology, broad ethical, government policy and economics, abortion, assessment of implemented service, and those that were not available in English. 

This process was independently conducted by two authors who reviewed article titles, article abstracts, and then full text to identify articles for inclusion. At each step, a third author resolved any disagreements. 

### 2.3. Review and Charting of Results

Full-text review of the selected studies was completed and then summarised into a table of six headings, including the author(s), country, year of publication; aim; study design; key outcomes; identified barriers; and recommendations or implications (Table 1). The studies were grouped into sections outlining whether the study investigated barriers to initiating, continuing ART or both. Most studies identified multiple contributing barriers, which were themed and then further refined in alignment with the Consolidated Framework for Implementation Research (CFIR) “Outer setting” and “Inner setting” domains of influence [13]. The frequency and refinement of barriers discussed in the identified studies are illustrated in Table 2. 

**Table 1 pharmacy-11-00017-t001:** Summary of included studies relevant to barriers to patients accessing or continuing ART.

Author/Year/Country	Aim	Study Design	Key Outcome	Identified Barriers	Recommendations/ Implications
**Barriers to Initiating ART**					
Eisenberg ML, et al. [14] 2010 USA (Northern California)	To document the rate a cohort of infertility patients declined to pursue treatment and to determine reasons for this decision.	Mixed Methods Study Questionnaires followed by interviews of patients attending clinic for infertility 434 participants	Of 434 in cohort 13% did not pursue any infertility treatment.	Increased age Lower education Lower socioeconomic status Lower financial means Psychological (Depression)	*Recommendation* Introduce methods to detect and treat depression at initial infertility evaluation.
Hammoud AO, et al. [15] 2009 USA	To characterise the demographic correlation of IVF availability and utilisation.	Cross-sectional demographic analysis of public data Patients undergoing IVF	Lower availability of IVF physicians in USA vs. other developed countries. Lack of IVF insurance coverage correlated with low utilisation rates.	Geographic location (Country/State/ Less urbanisation) Lower education Lower financial means Health insurance coverage (lack of or limited)	
Herbert DL, et al. [10] 2010 Australia	To identify early users of fertility treatment (<34 years) and associations between various barriers.	Cross-sectional survey Infertile women from fertility clinics and community who had and had not used hormones/IVF as treatment 290 participants	Women (<34 years) living in major cities and having private health insurance is associated with early use of treatment for infertility at clinics.	Geographic location (outside major cities) Health insurance coverage (nil)	*Implication* Inequity of service for infertile women.
Yudin MH, et al. [16] 2010 Canada	To estimate the types of services available in fertility clinics in Canada for HIV positive individuals and couples.	Survey Registered fertility clinics in Canada clinics surveyed to assess availability of services for infertility OR viral transmission risk reduction in achieving pregnancy 28 clinics	Access to fertility investigation and treatment in Canada is limited and regionally dependent particularly for HIV positive patients.	Geographic location (distance from clinic) HIV positive status	*Recommendation* Develop strategies to increase access to appropriate service.
Blanchfield B, et al. [17] 2015 USA	To determine whether racial and/or sexual minority people receive help to become pregnant at the same rate as those not in these minority groups.	Demographic and socioeconomic analysis of public data (National Survey of Family Growth) Women aged 21–44 7463 participants	Heterosexual White women receive medical fertility assistance at nearly double the rates of non-White, sexual minority, or both. Differences in rates of help only partially mediated by insurance coverage and income.	Race/culture/ethnicity (non-white) Health insurance coverage (nil) Lower financial means (lower income)	*Implication* Sexual minority less likely to have insurance.
Chin HB, et al. [18] 2015 USA	To examine the persistence of a racial disparity in visiting a doctor for help getting pregnant.	Data analysis of cohort study Women aged 22–45 1073 participants	Black women were less likely to visit a doctor to get help to attain pregnancy After reporting infertility white women waited a year and black women 2 before accessing treatment.	Race / culture / ethnicity (Black women) Psychological (social stigmatisation around infertility and disappointing spouse)	*Recommendation* Development of online resources and encouragement of initiation of conversation with primary health care provider.
Harris K, et al. [19] 2016 Australia	To investigate disparities in access to assisted reproductive technology based on socio-economic status and geographic remoteness.	Data analysis of three datasets Women who accessed ART from 2009–2012 85,602 participants	Most disadvantaged/2nd most advantaged socio-economic status quintile had a 16%/6% reduction in access compared with the most advantaged quintile. Living in regional and remote areas had a 12% reduction in access.	Geographic location (remoteness) Lower socioeconomic status	*Recommendation* Change to public health policies to reduce inequity of access.
Chin HB, et al. [20] 2017 USA (Georgia)	To determine if there are any differences by place of residence in visiting a doctor for help getting pregnant.	Mixed methods study (Data analysis of cohort study and interviews) Women aged 22–45 1073 participants	Women visiting doctor for help getting pregnant ranged from 13% in small town/rural to 17% in suburban areas. Small town/rural and small metropolitan more likely to receive medications, less likely to receive testing alone or IVF. IVF 20% urbanised, 15.2% suburban, 12.5% small metropolitan, 0% small town/rural.	Geographic location (non-suburban location) Psychological (embarrassment / stigma)	*Recommendation* Increased communication from care providers to patients about reproductive goals and infertility care.
Dimitriadis I, et al. [21] 2017 USA.	To examine whether race affects: duration of infertility prior to seeking evaluation, diagnosis, treatment cycle characteristics, and outcomes.	Retrospective cohort data analysis Data examined from 5186 IUI cycles was retrospectively reviewed 1495 participants	Time to infertility-Asians and Hispanics compared to Caucasians waited significantly longer prior to seeking fertility evaluation. No effect of race on the average number of cycles required to achieve clinical pregnancy.	Race/culture/ethnicity (Asian and Hispanic)	
Janitz AE, et al. [22] 2019 USA	To assess racial differences in utilisation of infertility services (emphasis on American Indian, Alaskan natives).	Secondary analysis of cross-sectional data from survey (NSFG) Women aged 15–44 1824 participants	Disparities accessing and reduced service utilisation observed American Indian/Alaska Natives, Hispanic, and Black women compared to White. Asian/Pacific Islander similar prevalence of using services to Whites, except a lower prevalence seeking advice.	Race/culture/ethnicity (American Indian/Alaska Natives, Hispanic, and Black)	*Recommendations* Study to understand racial discrepancies in access and utilisation
Insogna IG, et al. [23] 2020 USA	To test the hypothesis that under-represented minority women, including Hispanic/Latina and African American or Black women, will be more likely to report greater socioeconomic and cultural barriers to infertility care compared with white women.	Cross-sectional survey Women aged 18–44 242 participants	No significant differences in education level, insurance type, socioeconomic barriers in access to care. Hispanic/Latina less likely to know if insurance covered infertility treatment.	Cost (out of pocket) Health insurance coverage (knowledge of insurance related to culture)	*Recommendations* Public health messaging and education around how to gain financial coverage.
Kyei JM, et al. [24] Ghana 2020	To assess experiences of clients accessing ART services in Accra Ghana.	Qualitative study using Semi-structured interviews Men and women with infertility undergoing ART 12 women and 6 men	Five major challenges were identified at every phase of ART treatment: high cost, long distance to treatment centres, drug treatment challenges, disturbances in daily routine and work, anxiety around pregnancy outcome.	Geographic location (distance to treatment centres) Cost Psychological (Drug treatment challenges and anxiety around success of treatment) Multi-barrier (Change to routine)	*Recommendations* Counselling units to be added to centres, more insurance coverage.
Gilbert E, et al. [9] 2021 Australia (NT)	To explore health care provider (HCP) perspectives on the health burden of infertility among Aboriginal and Torres Strait Islander people, as well as factors that may affect access to infertility treatment for this group.	Qualitative study using semi-structured interviews Health care providers 12 participants– 8 doctors, 3 nurses, 1 aboriginal health practitioner	HCP perceive an underestimated health insurance in this patient population. Perceived barriers to accessing fertility care were reported.	Race/culture/ethnicity (Communication, language, fertility health literacy, shame and stigma, lack of culturally appropriate service) Geographic location (Distance from service) Psychological (shame and stigma)	*Recommendations* Increased patient education. Ongoing patient support Provision of culturally safe environment.
Koppen K, et al. [25] 2021 Germany	To examine the factors that assist and prevent individuals from seeking help for infertility and from using Medically Assisted Reproduction (MAR) in Germany.	Data analysis based on data from a longitudinal study Participants aged 25 or older who had tried to become pregnant or who were pregnant since previous interview 1446 participants	Utilisation of treatment was socially selective with married childless couples with a higher income rating the highest and younger couples with a less solid financial background, particularly those not married, appearing to face barriers to the use of medically assisted reproduction because of restrictive guidelines, corresponding legislation and limited insurance coverage.	Increased age Lower financial means Health insurance coverage (nil or limited) Marital status (not married)	
Barriers to continuing ART					
Bedrick BS, et al. [26] 2019 USA	To investigate factors involved with early IVF treatment discontinuation.	Secondary analysis of a retrospective cohort study Women undergoing IVF (first attempt) and did not achieve live birth 699 participants	Women without IVF insurance coverage had 3 times higher odds of treatment discontinuation. African American 3 times the odds of treatment discontinuation than White. Poor prognosis associated with greater likelihood discontinuing treatment.	Race / culture / ethnicity (African American) Health insurance coverage (nil) Confidence in fertility treatment (poor prognosis)	
Domar AD, et al. [27] 2010 USA	To determine the primary reason why insured patients, drop out of IVF treatment in the United States and to identify methods to decrease such behaviour.	Prospective patient survey Women age <40 with insurance for at least 3 cycles, did not conceive and did not return for third cycle 132 participants	Treatment termination was most commonly due to stress and psychological issues.	Geographic location (distance away from service) Psychological (stress, depression, relationship issues) Lower financial means Medication side effects	*Recommendations* Provide information on how to deal with psychological issues and immediate access to psychological services.
Domar AD, et al. [28] 2018 USA	To study the reason(s) why insured patients discontinue in vitro fertilisation (IVF) before achieving a live birth.	Cross-sectional survey study Women who have completed one IVF cycle but did not return within a year and no live birth 383 participants	Discontinuation was reported to be contributed to due to stress, financial burden or conceiving spontaneously.	Geographical location (distance from service) Lower financial means Health insurance coverage (lost) Psychological (stress) Medication side effects Confidence in fertility treatment (dissatisfaction / decreased confidence in provider)	*Recommendations* Investigate psychological interventions.
Maxwell E, et al. [8] 2018 Canada (Newfoundland)	To explore how barriers to accessing fertility services affect the treatment decisions made by fertility patients and service providers in Newfoundland and Labrador.	Qualitative study using semi-structured interviews ART patients and ART service providers 11 patients and eight service providers	Patients and providers make treatment choices to maximise likelihood of pregnancy success and increase accessibility (and costs) which can result in less effective care and sometimes potential risk to the patient.	Geographic location (isolation, number, and location of services available, partner separation) Cost Psychological (social stigma)	*Recommendations* Provide teleconsultations to make fertility care more accessible in rural and remote areas of the province.
Barriers to initiating and continuing ART					
Bennett LR, et al. [29] 2012 Indonesia	To improve understanding of infertility patients’ health seeking behaviour and patterns of access to infertility treatment in Indonesia, and to highlight possibilities for improving access.	Interviewer administered survey Female infertility treatment patients (from 3 Indonesian infertility clinics) 212 participants	Patients identified various barriers to accessing ART including: low confidence in fertility treatment; number and location of clinics; lack of well-established referral system; cost of treatment; fear of diagnosis of sterility; vaginal examination embarrassment. Women’s age of marriage and the timing of their initial presentation to gynaecologist were NOT found to be barriers to timely access to infertility care.	Geographic location (travel away from service) Cost Psychological (fear and embarrassment (shame)) Confidence in fertility centre (low)	*Recommendations* Increased patient education. Increased resources to reduce travel. Improve financial accessibility. Expansion of referral system.

**Table 2 pharmacy-11-00017-t002:** Summary of barriers identified by frequency.

Barrier	Initiating/Continuing	Number of Studies	Description of Barrier
Geographic location [8,9,10,15,16,19,20,24,28,29]	Initiating Continuing	11	Outside major cities Less urbanisation Geographic remoteness Embarrassment/shame with infertility related to geographic location Distance to service Travelling time and cost Inconvenience Away from emotional support Separation from partner Accessibility to IVF centre
Psychological [8,9,14,18,20,24,28,29]	Initiating Continuing	9	Depression Anxiety around success of treatment Social stigma Embarrassment/shame
Medication side effects [27,28]	Continuing	2	
Confidence in fertility treatment [26,28,29]	Continuing	3	Dissatisfaction with provider Confidence in provider
Race/culture/ethnicity [9,17,18,21,22,26]	Initiating	6	Non-white race Cultural relation to embarrassment/shame/stigma Communication and language Culturally appropriate service
Marital status [25]	Initiating	1	
HIV positive status [16]	Initiating	1	Safe conception
Socioeconomic status [14,19]	Initiating	2	
Health insurance coverage [10,14,15,17,25,27,28]	Initiating	7	Coverage by health insurance Classification of infertility as not a health condition but as a socially constructed need Knowledge of cover
Disposable income [14,15,17,25,27,28]	Initiating Continuing	6	Available funds Lower median income
Cost [8,23,24,29]	Initiating Continuing	4	
Lower education level [9,14,15]	Initiating	3	Formal education Health literacy
Increased age [14,25]	Initiating	2	

## 3. Results

### 3.1. Selected Studies

Figure 1 illustrates the PRISMA (Preferred Reporting Items for Systematic Reviews and Meta-Analyses) 2020 flow diagram for scoping review strategy [30]. The original database searches produced 4599 records. A total of 1609 duplicates were removed, leaving 2990 studies to screen. These 2990 studies were screened based on the title and contents of the abstract, with 2964 excluded based on the exclusion criteria. The full text of the 26 remaining studies was sought for retrieval, with 2 records not retrieved due to the full text not being available in the English language. A review of the full text of the 24 remaining studies was completed, with 19 remaining studies suitable for this review (Figure 1). The studies published ranged in date from 2009 to 2021. While many of the studies were published in the United States (11), studies were also based in Australia (3), Canada (2), Indonesia (1), Germany (1), and Ghana (1) [8,9,10,14,15,16,17,18,19,20,21,22,23,24,25,26,27,28,29].

The studies investigated barriers to people either initiating or continuing infertility treatment or both. Fourteen of the studies focused on barriers to initiating ART, with barriers to continuing treatment being the focus of 4 studies [8,9,10,14,15,16,17,18,19,20,21,22,23,24,25,26,27,28]. One study discussed both initiating and continuing fertility treatment [29]. The general studies investigated reasons why a person or couple were less likely to initiate and/or continue ART, characterised demographic availability and utilisation, or considered reasons why people are less likely to initiate or continue ART. Eleven of the 19 studies were in this category [8,9,10,14,16,24,25,26,27,28,29]. The remaining eight studies investigated specific health determinants such as geographic location, a patient being of a racial or sexual minority, and/or socioeconomic status. A quarter of the studies specifically investigated race as a potential barrier to ART [9,17,18,21,22,26]. 

The methods used varied, with the most prominent being the retrospective analysis of datasets and large cohort surveys [15,17,18,19,20,21,22,25]. Five of the studies used surveys, with 60% surveying patients and the remaining directed towards the fertility clinics themselves [10,16,23,27,29]. 

### 3.2. Barriers to ART

Barriers to accessing or continuing ART identified from each of the studies consisted of non-modifiable, social and economic, educational, and environmental determinants of health [31]. The most mentioned barrier was that of geographical location, which was highlighted in 11 of the 19 studies [8,9,10,15,16,19,20,24,27,28,29]. The second most common barrier appearing in half of the studies centred around the psychological barriers to initiating or continuing ART [8,9,14,18,20,24,27,28,29]. Health insurance was also identified in 37% of studies and included no insurance, no coverage for ART, and a lack of knowledge of insurance coverage [10,15,17,23,25,26]. A third of the studies discussed at least one of the following barriers of lower financial means, health insurance coverage, psychological, and race/culture/ethnicity [9,10,14,15,17,20,21,22,23,24,25,26,27,28]. The barriers of increased age, lower education level, lower socioeconomic status, cost, confidence in fertility treatment, and medication side effects were all identified in more than one study [14,16,19,24,25,27,28,29]. Marital status and HIV-positive status were referred to in one study each [16,25]. All studies that discussed or identified the impact of socioeconomic status, minority women and culture or ethnicity reported that those categories were considered a barrier to initiating or continuing ART, except in one study that determined they had no impact. Table 2 provides a summary of the frequency with that barriers were identified and/or discussed in this scoping review.

#### 3.2.1. Geographical Location of Patient

Geographical location, the most frequently identified and investigated barrier to a patient accessing ART, can act as a barrier in several ways, ranging from the state or country that a patient lives in, and the number and location of fertility clinics, to geographic isolation [29]. It was identified that worldwide, most ART services are in metropolitan or larger regional areas, which can make distance an issue for people who live outside of these localities [7,10,15]. Due to the nature of most ART services, patients are often required to be at the clinic at specific times for investigation or intervention, which is required for the success of the treatment [24,28]. Therefore, it is not surprising that Hammoud et al. found that ease of accessibility to ART service providers and patient utilisation of IVF were highly correlated [2]. 

Different health insurance and financial models in various countries highlighted the discrepancy in accessibility which is not only limited to how many services are available but also the criteria for accessing these services [15]. Hammoud et al. found that compared to Western Europe and Australia, the United States had a significantly lower utilisation and accessibility to IVF treatment [15].

The availability of infertility treatment facilities in less-populated locations was also identified as a barrier due to potential wait times and a lack of choice in service. As a result of geographic isolation from ART service providers, it was established that patients living a distance away from services had costs above the usual expense of ART [9,19,24,29]. Increased expenditure was associated with travel and accommodation, time away from work (due to the need to relocate), and partners’ time away from work [9,24]. One study conducted in Ghana particularly mentions the costs associated with patients living a distance away from providers [24]. It is noted that the cost of infertility treatment alone is expensive; however, when the added costs of travel, accommodation and time away from work are combined, the process is "very expensive" [24]. Similar concerns were highlighted in a study conducted in Australia, which sought the views of health professionals in the Northern Territory [9]. The cost of treatment and travel as a barrier was thought to be compounded due to the stress of being away from a familiar environment and the emotional support provided by family and the community [9]. 

Geographical location was found not only to contribute to whether a patient accessed treatment but also the type of treatment offered and/or selected, with women from small towns and rural areas more likely than those from urbanised or suburban counties to receive oral medications rather than more effective IVF to treat infertility [20,22].

#### 3.2.2. Psychological Barriers

The psychological status and effects on the patient were shown to act as a barrier to both initiating and continuing ART. Existing depression was identified as a factor in decreasing the likelihood of subjects pursuing infertility treatment [14]. Initiating treatment was also hindered due to patient experiences of stigma, shame, and embarrassment, which to a lesser extent also contributed to discontinuation of treatment [9,18,23,29]. This was highlighted universally but was particularly related to race, culture, and urbanisation [9,18,23,29].

A lower level of urbanisation (independent of the race or culture of a patient) was also correlated with an increased level of shame and stigma associated with infertility and receiving treatment [20,23]. One study from Georgia in the United States identified that women living in small towns and/or rural counties reported less comfort with ART but greater comfort with adoption [20]. Anxiety around the success of the treatment, the effect on the patient’s relationship and concerns about the treatment itself contributed to barriers to both initiating and continuing treatment [14,27].

Continuing ART was affected by the confidence that patients had in fertility treatment. A study conducted in Indonesia showed high rates of patients switching between providers due to perceived treatment failure [29]. Similarly, Domar et al. in the United States, indicated that patients discontinued a treatment centre after no success to seek a second opinion when they were not satisfied with the initial care given or due to hearing good things about another centre [28].

#### 3.2.3. Minority Group Barriers

As has been alluded to in previous paragraphs, race, culture, and ethnicity is indicated to act as a barrier to someone initiating or continuing ART [17,18,21,22,23,26]. For the most part, barriers to accessing infertility treatment were found to be increased amongst non-Caucasian people and were believed to be attributed to cultural beliefs, communication, and a lack of culturally appropriate services [17,18,21,22,23,26].

Minority women were investigated in six of the 19 studies and were classified as minorities based predominantly on race/ethnicity, but sexual orientation was also investigated in two of the studies [9,17,21,22,23,25,26]. Barriers to accessing ART in this group were primarily financial due to the patient’s inability to access insurance coverage for ART when conventional infertility was not diagnosed [17]. Same-sex couples in Germany, however, faced different barriers to accessing fertility treatment, and this is not only limited to same-sex couples [25]. Marital status as an impediment to accessing ART only directly appeared in one study from Germany [25]. Legal guidelines and insurance coverage rules in Germany outlined that medical-assisted reproductive (MAR) therapy is primarily granted to married couples, with non-married cohabiting couples 40% less likely to use MAR [25]. Koppen et al. determined that the marital status effect was independent of other sociodemographic variables and identified Germany to be particularly strict in the regulation of marital status [25].

HIV-positive couples (male, female, or both), were only reported on in one study in Canada, which discussed the barriers at length [16]. The need for ART in this group centred around the safe conception of a child to prevent transmission of HIV to the other partner or to the child [16]. Overall, access to fertility investigation and treatment in Canada was deemed limited and regionally dependent, even more so for the needs of HIV couples [16].

#### 3.2.4. Financial Barriers

The financial situation of a patient was identified as a barrier to patients accessing ART. Being of lower socioeconomic status, having a lower median income, and having no or little coverage by health insurance were all found to act as barriers to initiating or continuing treatment [8,9,10,14,15,17,23,24,25,27,28,29]. Health education was also integrated into this barrier, with unawareness of insurance coverage (and hence affordability) discouraging some patients from accessing treatment [23]. This was particularly evident in the United States, where patients were unaware of mandated insurance coverage for infertility treatment and/or their insurance coverage [23].

Insurance coverage and the classification of infertility as a health condition versus a socially constructed need varied between states and countries due to various medical support systems in place. Hammoud et al. determined from a data analysis that IVF availability and utilisation were higher in states with IVF insurance coverage [15]. Significant out-of-pocket expenses contributed to inequity to access, dependent on disposable funds and the willingness of patients to spend money on treating infertility [15].

#### 3.2.5. Education Level of Patient

All studies that identified a lower educational level of the patient acting as a specific barrier to accessing ART were based in the United States [15]. Eisenberg et al. found that women who had less than a college education had a 79% higher chance of not pursuing treatment compared with women who had a college education [14]. Although limited information was provided on the educational level of patients, the information suggested that it may contribute to a person not only accessing ART but the type of ART accessed [14,15].

#### 3.2.6. Age Barriers

Non-modifiable issues such as age were the least mentioned barrier. Eisenberg et al. determined that for each 5-year increase in a woman’s age at initial reproductive endocrinology evaluation, the odds of not pursuing treatment increased by 77% [14]. In Koppen et al., a study based in Germany, access to ART significantly decreased in women over the age of 40 [25].

## 4. Discussion

Nineteen studies published between 2009 to 2021 contributed to this scoping review. The overall aim was to determine the current international evidence for barriers to accessing ART. Once barriers were identified, this review’s goal included determining how pharmacists could address identified barriers and improve access and outcomes for ART patients, particularly those in vulnerable and marginalised groups in rural and remote areas.

Broad themes of potential enablers and barriers to ART, including patient characteristics, location and availability of ART services, and cost of ART, were initially determined, which were further refined according to the Consolidated Framework for Implementation Research (CFIR) [13,32,33]. This review indicates that barriers exist to both initiating ART and continuing ART (once started). The themes were further broken down into subthemes considering the CFIR "Outer Setting" domain of influence, which included geographical location or level of urbanisation, ethnicity, psychological status, educational level of the patient, and the external policies and incentives that influence financial affordability. The CFIR "Inner Setting" domain of influence included cultural influences and the need for culturally appropriate accessible ART services [13,32,33]. For the most part, the same barriers existed to both initiating and continuing ART. However, some differences were identified.

Patient characteristics and location of residence comprised the most identified barriers to both initiating and continuing ART. The location of the residence potentially impacted accessibility in several ways. From the country of residence through to the degree of urbanisation, evidence of relationships between cost, socioeconomic status, psychological effects, and accessibility to ART services was illustrated [8,9,10,15,19,20,24,27,28,29]. The imbalance in geographical access to ART services was recorded as being a result of sparsely populated areas not being financially viable to service [8]. Patient’s travelling an increased distance to access services was not only inconvenient but also contributed to financial and psychological stress, potentially acting as an additive barrier to accessing ART [8,9,10,15,16,19,20,24,27,28,29]. The degree of urbanisation also influences the choice that patients have when accessing services, with more choices of clinics available in larger cities [16,20]. The type and potential effectiveness of treatment offered to patients could also be attributed to the location of the patient, with less invasive but also fewer effective options offered to reduce cost and the inconvenience of travel [20,22].

Existing psychological conditions can act as a barrier to initiating ART as well as being exacerbated or generated because of going through the strenuous ART process [14]. Although not directly classified as a psychological condition, the shame and stigma associated with being infertile and having the need for medical intervention contribute to psychological symptoms and can act as a barrier to accessing ART, predominantly to initiating [8,9,20]. Shame and stigma associated with infertility and treatment for such were associated with people who live in less urbanised areas and those of particular cultures and ethnicity [8,9,20]. Patient stress has also been identified around concerns about using medications for ART. The cost of medications to the patient and the potential ramifications of user error contributed to the stress and psychological impacts on patients [27,34,35]. A previous study has highlighted the potentially increased role of pharmacists by illustrating improved outcomes for patients who have experienced counselling from a pharmacist for ART medications, with a 29.3% increase in knowledge [36].

Ethnicity, culture, sexual orientation and/or identification, and marital status were found to be barriers in certain aspects of accessing ART. It was found that non-Caucasian individuals, in general, were less likely to seek advice for infertility and certain races were less likely to access treatment [17,20,21,22]. Reasons are likely to be complicated and varied with more in-depth analysis into this area required if further understanding is to be attained.

For the most part, those of sexual minority sought fertility treatment to enable conception as opposed to treatment of a traditional diagnosis of infertility, which influenced the availability of insurance support [17,25]. Broadly, the lack of financial coverage explained the barrier to sexual minority patients accessing ART. Similarly, for HIV patients, sourcing help to conceive was predominantly to help reduce HIV transmission rather than help a diagnosis of infertility [16]. Although a non-married person is not considered to be a ‘minority’, what is highlighted is that different rules, regulations, and financial assistance may be available in these cases, which can influence accessibility [25]. Increasing cultural competency in pharmacy practice has the potential to reduce health disparities and be expanded upon in relation to other minority groups [37,38]. Addressing the needs of diverse people in the community can improve access to health information and effective and safe use of medications in an inclusive and respectful manner [37,38].

The cost and expense of ART are well known, which in general, means that those with a higher disposable income are more likely to be able to afford ART. This alone though does not explain the entirety of financial barriers to accessing ART. A patient’s insurance coverage, subsidisation by the government, geographical location, socioeconomic status, and a patient’s acceptability of spending significant money to attempt pregnancy all contribute. Health providers were shown not always to be aware of the costs and rebates available, increasing the difficulty for patients to be educated about associated costs and make informed decisions [9].

The educational level that an individual had achieved was shown to impact the likelihood of a patient accessing ART [14,15]. Higher education level attainment is associated with increased financial means as well as the ability to be informed about infertility, accurate source information regarding treatment and access to treatment [14,15]. Education programs and health promotion around infertility were suggested to improve health literacy and to enable awareness of fertility status, sources of information, and access to medical help if required [14,15]. Pharmacists’ accessibility enables them to have a key role in increasing patient knowledge of infertility, modifiable risk factors, and available treatments to address the barrier of education [39,40,41].

Multiple and various barriers were identified, with some studies contributing recommendations to aid in the breakdown of inequity of access to ART. On a large scale, directly addressing financial barriers and increasing government support in terms of subsidies and services could act as an enabler to increasing the ability of patients to access ART [19,24,29]. From a pharmacist and health professional perspective, health education and health promotion to increase awareness around infertility and identify at-risk groups is a potential strategy to increase access to ART services in a timely manner and to contribute to the normalisation of this topic, decreasing stigma [9,22]. Cultural education of nurses, doctors, pharmacists, and allied health professionals is important to address stigma and shame, enabling the provision of care in a culturally sensitive and appropriate way [37,38]. Other options suggested to address geographical barriers are the introduction of satellite clinics and telehealth services utilising the accessibility of local health professionals, including pharmacists [8,16,18]. The potential exists for an expanded practice of health professionals to enable pathology tests, ultrasounds etc., to be done at a time and location that is more suitable to the patient [42]. Screening for psychological conditions such as depression and the appropriate referral and treatment of such could be a way to address this barrier and support the patient both mentally and physically, with a multidisciplinary approach, including pharmacists, suggested as a potential way to address these barriers [43,44].

As medications are at the forefront of ART, the role of the pharmacist as a provider of medications can be seen to be an important aspect of this service; however, studies investigating the pharmacist’s role are limited. A small number of studies have identified that there are specialist pharmacists that work with ART. However, it is not an officially recognised role and lacks formal training requirements and qualifications [34,45]. Pharmacists not considered ART specialists do, however, still work with and care for ART patients. Patient experiences between specialist and non-specialist pharmacies were reported in a study that found that patients receiving ART medications were more satisfied with the service provided by "specialist" pharmacists when compared with "non-specialists" [34].

Pharmacist involvement with patients around fertility advice, medication management, psychological screening, and cultural safety could alleviate some of the stress and distress associated with ART [36,43,44,46,47,48]. Accessibility of pharmacists is high, and the profession is well situated to address, identify, and educate patients on the topic of infertility and the services available and to support patients throughout their journey, especially regarding medications [34,36,41,45]. Psychological concerns have the potential to be identified and addressed through screening tools, listening emotional support, and referral. In rural and remote areas, pharmacists could aid in decreasing communication and "distance" barriers between ART service providers and patients through the implementation and facilitation of telehealth and networking with ART clinics [36,43,44,46,47,48,49,50,51,52].

## 5. Strengths and Limitations

This scoping review identified barriers to accessing ART globally. The review has taken into consideration both modifiable and non-modifiable determinants to identify ways to reduce these identified barriers. The inclusion of studies from American, Canadian, Australian, Asian, European, and African origins allowed for a good representation globally. Limitations to this review included the inability to review non-English studies and having most of the studies based in the United States (11 of 19 studies), limiting the generalizability of the findings.

## 6. Conclusions

There is an inequity of access to ART, with barriers existing that prevent those affected by infertility from accessing services to improve their chances of a successful pregnancy. These barriers are often multifactorial and interlinked. Infertility is not always classified as a medical condition but rather a socially constructed need, with infertility services typically located in large regional and metropolitan cities [7,8,9,10,15]. Whereas the cost of treatment is certainly an identified barrier to accessing ART, the geographical isolation of patients from fertility treatment services was the most prominent barrier identified in this scoping review [8,9,10,15,16,19,20,24,27,28,29]. For those living in rural and remote areas, ART services being located too far from home is a deterrent, not just time and inconvenience, but also contributes to an increase in financial, social, and psychological burden [8,9,14,18,19,20,24,27,28,29]. This in turn impacts the likelihood of people living in these areas initiating or continuing with infertility treatment and/or the type of treatment options used. The clarity in the pharmacists’ role in ART, the knowledge required and training in this area of pharmacy practice has the potential to benefit patients and address several identified barriers to patients accessing ART. For all health professionals, understanding the people in their community, their unique needs, cultures, and ways of communicating, is beneficial to breaking down barriers to accessing medical treatment, including ART.

## Figures and Tables

**Figure 1 pharmacy-11-00017-f001:**
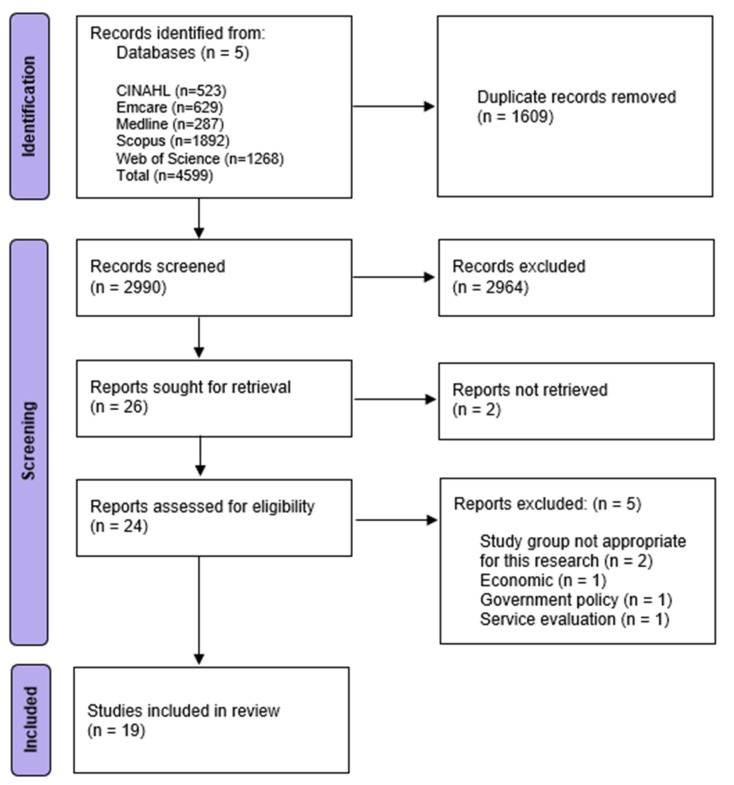
PRISMA Flow Diagram for scoping review search strategy [30].

## Data Availability

Not applicable.

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
