# Peer review of "Inequity of Access: Scoping the Barriers to Assisted Reproductive Technologies"

_pharmacy, 2023, doi:10.3390/pharmacy11010017_

Round 1
Reviewer 1 Report
I believe this was a very well conducted and written scoping review. I felt the purpose of the review was clear aiming to explore barriers to accessing assisted reproductive technologies (ART). Additionally, I liked that this was taken one step further and incorporated information into the discussion section regarding the potential role that pharmacists could have in addressing these barriers. I think the main contribution of this review is that it presented a wide range of information on the topic and many different barriers that were identified. This was well organized within the results section through both the tables and text. Although not a direct contribution of the work a strength is that is alludes to future areas for interventions and research to address the barriers identified.
The review conducted appeared to be complete and include all of the pertinent studies on the topic, but was still specific enough to be able to present good themes and draw conclusions from the information synthesized. Fertility issues and ART is a growing area of healthcare, so I believe this topic is of good relevance as it is becoming more prevalent. The international scope of the studies included which highlighted some similarities and differences based on study location provided interesting insights on this topic. All citations for both the articles included in the review as well as other references seems recent and relevant.
Specific Comments on Strengths
Methods – Good description of inclusion and exclusion criteria for studies.
Results – Nice PRISMA Flow Diagram and brief description of types of articles included prior to more in-depth analysis. Text in Section 3.2 and Table 2 complement each other well and provide good summary of trends/themes across different studies.
Discussion – Highlights links and associations between different types of barriers. Section was very well written and incorporated additional literature about what could be done to address the barriers identified, specifically the role the pharmacist could play in this.
Specific Comments on Improvements
Results – Table 2 cites very specific barriers in the left-hand column. The text in subsections 3.2.1 – 3.2.6 seems to group the specific barriers more broadly from the table into more comprehensive themes/domains. For example, 3.2.4 discusses financial barriers, which includes the specific barriers from the table of socioeconomic status, disposable income, cost, and health insurance coverage. A potential suggestion would be to reformat the table to be present the barriers in order of discussion in the text or grouping them by the larger theme.
Reviewer 2 Report
I think this is a valuable review to inform researchers and clinicians about the barriers to accessing and continuing assisted reproduction.
I recognize that is is challenging to conduct a scoping review on this topic as the true barriers are often unreported in many clinical studies. However, I think this scoping review provides some insights.
I find the title of the manuscript is rather leading as the authors are already suggesting that distance is the issue. I am not entirely convinced that geographic distance to clinics is an important enough barrier to warrant including it in the title of the manuscript. Whilst the authors suggest that distance was the main barrier mentioned in the studies identified, some of these factors could be construed as economic as the socioeconomic and fertility patterns and rural and urban areas are likely to be different. I would like to see a more generic title for a review of this nature.
